# Rabies Virus Regulates Inflammatory Response in BV-2 Cells through Activation of Myd88 and NF-κB Signaling Pathways via TLR7

**DOI:** 10.3390/ijms25179144

**Published:** 2024-08-23

**Authors:** Yuan Xie, Yinglin Chi, Xiaoyan Tao, Pengcheng Yu, Qian Liu, Minghui Zhang, Nuo Yang, Shuqing Liu, Wuyang Zhu

**Affiliations:** Key Laboratory of Medical Virology, Ministry of Health, National Institute for Viral Disease Control and Prevention, NHC Key Laboratory of Biosafety, Chinese Center for Disease Control and Prevention, Beijing 102206, China; 201721490023@mail.bnu.edu.cn (Y.X.); chiyinglin1998@126.com (Y.C.); txy212@126.com (X.T.); ypccz@163.com (P.Y.); liuq@ivdc.chinacdc.cn (Q.L.); zhangmh@ivdc.chinacdc.cn (M.Z.); 13513228620@163.com (N.Y.)

**Keywords:** rabies virus, TLR7, Myd88, NF-κB pathway

## Abstract

Rabies is a fatal neurological infectious disease caused by rabies virus (RABV), which invades the central nervous system (CNS). RABV with varying virulence regulates chemokine expression, and the mechanisms of signaling pathway activation remains to be elucidated. The relationship between Toll-like receptors (TLRs) and immune response induced by RABV has not been fully clarified. Here, we investigated the role of TLR7 in the immune response induced by RABV, and one-way analysis of variance (ANOVA) was employed to evaluate the data. We found that different RABV strains (SC16, HN10, CVS-11) significantly increased CCL2, CXCL10 and IL-6 production. Blocking assays indicated that the TLR7 inhibitor reduced the expression of CCL2, CXCL10 and IL-6 (*p* < 0.01). The activation of the Myd88 pathway in BV-2 cells stimulated by RABV was TLR7-dependent, whereas the inhibition of Myd88 activity reduced the expression of CCL2, CXCL10 and IL-6 (*p* < 0.01). Meanwhile, the RABV stimulation of BV-2 cells resulted in TRL7-mediated activation of NF-κB and induced the nuclear translocation of NF-κB p65. CCL2, CXCL10 and IL-6 release was attenuated by the specific NF-κB inhibitor used (*p* < 0.01). The findings above demonstrate that RABV-induced expression of CCL2, CXCL10 and IL-6 involves Myd88 and NF-κB pathways via the TLR7 signal.

## 1. Introduction

Rabies is a fatal zoonotic disease characterized by hydrophobia, wind phobia, pharyngeal muscle spasm [1,2]. Owing to constraints in medical resources and suboptimal vaccine administration, more than 60,000 individuals succumb to rabies worldwide each year [3,4]. RABV is the causative agent of rabies, and also affects the CNS in a variety of mammalian species [3]. RABV mainly accumulates in the peripheral area and migrates to the CNS via the motor or sensory nerves, eventually leading to the dysfunction of the CNS [5].

Pattern recognition receptors (PRRs) constitute a category of molecular sensors predominantly localized on the membrane of innate immune cells, capable of identifying one or several pathogen-associated molecular patterns (PAMPs) [6,7,8]. PRRs are classified into TLR, nucleotide oligomerization domain-like receptor (NLR) and retinoic acid-inducible gene I-like receptor (RLR), according to the differences in structure between the ligand binding domain and signal transduction domain. PRRs that can recognize PAMPs mainly include TLRs and RLRs with RABV infection. TLRs are a class of transmembrane receptor proteins, belonging to type I membrane proteins, which are mainly distributed in cell compartments or the endosomal membrane of monocytes and recognize extracellular PAMPs in cells [9,10,11]. The subsequent activation of signal transduction can lead to the release of inflammatory mediators, which play an important role in natural immune defense. Therefore, TLRs are considered to be a key regulatory element in the transformation of innate immunity to acquired immunity, and have come to be pathogen recognition receptors that have received extensive attention in recent years [12,13,14].

Viruses with different characteristics are recognized by different TLRs to trigger antiviral innate immunity. Signaling pathways associated with TLRs are primarily orchestrated by the adaptor proteins Myd88, TRIF and TRAM. These adaptor molecules facilitate the interaction between TLRs and their ligands, subsequently activating transcription factors such as NF-κB, which then trigger the expression of various cytokines and chemokines [15]. It has been confirmed that TLR3 was unique in utilizing TRIF without Myd88 [16]. TLR7 was mainly distributed in the endosomes of plasmacytoid dendritic cells (pDCs), macrophages, B cells and other cells. After TLR7 was bound to its ligand, it recruited the downstream connector molecule Myd88 [17,18,19]. It has been observed that TLR7 could recognize a variety of RNA viruses, including influenza virus, West Nile virus, HIV, yellow fever virus, lymphocytic choriomeningitis virus and murine pneumonia virus [19,20,21,22,23,24]. Furthermore, studies have shown that the receptor agonist of TLR7 could treat genital warts caused by human papillomavirus by enhancing antigen presentation and promoting antigen-specific Th1 cell-mediated immune responses [25]. However, Liu et al. found that antibody levels were elevated by using a TLR7 agonist as a vaccine adjuvant for RABV, suggesting that TLR7 played a critical role in RABV-induced neutralizing antibody production [26]. Li et al. reported that the pathogenicity of RABV was significantly increased in TLR7-deficient mice, but its specific effect and mechanism remained unclear [27]. These findings indicate that TLR7 could serve as a critical pattern recognition receptor for RABV, necessitating further investigation to elucidate its function in RABV pathogenesis. The study indicate that the accumulation of cytokines and chemokines could regulate both innate and adaptive immune responses to RABV infection [28]. The research demonstrated a marked upregulation in chemokine expression in mouse brain tissues infected with RABV, including CCL2, CXCL1, CCL5 and CXCL10 [29]. CCL2 and CXCL10 play a critical role in regulating the inflammation associated with RABV. CXCL10, predominantly expressed by neurons and microglia, was recognized as one of the earliest chemokines and exerts a positive effect during initial RABV infection [30,31]. Preliminary research indicates that CCL5 is critically involved in the neuroinflammatory processes during the infection of RABV, also promoting the apoptosis of T lymphocytes and microglia cells [3]. Additionally, a study indicated that following infection with RABV, there was an upregulation in cytokine expression (IL-1α, IL-1β, IL-6 and IL-10), concurrently accompanied by the recruitment of inflammatory cells into the CNS to facilitate viral clearance [32].

Our earlier findings indicate that RABV strains can activate various inflammatory responses through NF-κB signaling pathways in BV-2 cells, and NF-κB signaling pathways played a critical role in the expression of inflammatory agents of immune response [33]. Therefore, we aimed to explore whether or not NF-κB relevant signaling pathways are regulated by TLR7. Here, we examined the intracellular signaling mechanisms in BV-2 cells following infection with laboratory-adapted rabies virus strains (CVS-11). Then, the activation of relevant signaling pathways was detected via the silencing and over-expression of TLR7, and the expression differences of cytokines and chemokines were detected. Herein, we found that TLR7 was involved in the infection process of RABV, and that the TLR7/Myd88/NF-κB signaling pathway was activated. In addition, we also found that the expressions of CCL2, CXCL10 and IL-6 were significantly increased in the terminal stage of infection with RABV. These results elucidate a previously unrecognized immunoregulatory pathway involving TLR7 in BV-2 cells subjected to RABV infection. Furthermore, this investigation may serve as a foundational reference and provide essential experimental insights for the clinical management and prophylaxis of rabies. 

## 2. Results

### 2.1. RABV Exhibiting Varying Virulence Levels Provoke an Upregulation in TLR7 in Mouse Brain Tissues

TLR7 plays a critical role in RABV-induced neutralizing antibody production. To investigate the role of TLR7 in modulating the inflammatory response during RABV infection, we assessed TLR7 expression levels in brain tissue sections of mice infected with RABV exhibiting varying degrees of virulence through immunohistochemical assays. The results showed that the levels of TLR7 signals in mouse brain tissues were increased after RABV infection (Figure 1A,B). In addition, TLR7 signals in brain sections of mice infected with CVS-11 were stronger compared to SC16 and HN10 signals (Figure 1A,B). The data above suggest that the expression of TLR7 was increased after RABV infection and that the street strain showed weaker TLR7 signals.

### 2.2. RABV Induces Chemokine Expression In Vitro in a TLR7-Dependent Manner

A previous study demonstrated that the expression of proinflammatory chemokines including CXCL10, IL-6 and IL-27 was significantly elevated in the presence of TLR7 during RABV infection [34]. Our study showed that CCL2, CXCL10 and IL-6 were upregulated with RABV infection, and the expressions of CCL2, CXCL10 and IL-6 in BV-2 cells were higher than those in N2a cells [33]. To identify whether or not TLR7 was involved in RABV infection in N2a and BV-2, we initially detected the expression of TLR7 in N2a and BV-2 cells. The results showed that the expression of TLR7 was equivalent in N2a and BV-2 cells (Figure 2A,B). Meanwhile, to assess the effect of RABV on the activation of innate immune cells, the ability of RABV to stimulate the expression of CCL2, CXCL10 and IL-6 was examined in BV-2 cells. BV-2 cells were subjected to infection with RABV strains of varying pathogenicity (SC16, HN10 and CVS-11) at multiplicities of infection (MOIs) of 0.1, 0.5, 1, and 10. Then, proinflammatory chemokines in the culture supernatants were measured via ELISA after incubation for 24 h. As previously noted, we observed a substantial upregulation in the expression levels of CCL2, CXCL10 and IL-6 relative to those of the control group, indicating a dose-dependent relationship that peaked at an MOI of 1 (Figure 2A–C). In addition, the expression of CCL2, CXCL10 and IL-6 represented a non-significant difference with the infection of SC16, HN10 and CVS-11 (Figure 2A–C). Therefore, CVS-11 at an MOI of 1 was selected to infect BV-2 cells in subsequent experiments. In addition, to identify whether or not TLR7 was involved in RABV infection, BV-2 cells were pre-treated with or without a TLR7 inhibitor (HY-124603) for 2 h with the infection of CVS-11. We found that the blockade of TLR7 signals significantly decreased the production of CCL2 (72.42%, *p* < 0.01), CXCL10 (73.24%, *p* < 0.01) and IL-6 (76.32%, *p* < 0.01) (Figure 2D–F). The data above confirm that proinflammatory chemokine production induced by RABV was mediated by TLR7 signals.

### 2.3. Activation of Myd88 Pathway Regulates RABV-Induced Cytokine Expression via TLR7 Signal

The Myd88 signaling pathway was associated with the expression of cytokines, as a preliminary study has reported [35]. Therefore, to examine the role of the Myd88 signaling pathway in RABV infection, we measured the TLR7 signal, that of the downstream connector molecule Myd88, and those of IRAK4 and TRAF6 with CVS-11 infection at 30 min, 1 h, 2 h, 4 h, 6 h, 8 h, 12 h and 24 h via Western blotting. The results showed that TLR7, Myd88, IRAK4 and TRAF6 were unchanged until 6 h, an then their expressions showed an upward trend up until 24 h (Figure 3A,C). To evaluate whether or not RABV activated the Myd88 pathway through the TLR7 signal, BV-2 cells were pre-treated with or without the TLR7 inhibitor and agonist for 2 h. After incubation for another 6 h, we found that the blockade of TLR7 significantly reduced the expression of TRL7, Myd88, IRAK4 and TRAF6. In contrast, the overexpression of TLR7 compensated for the downward trend caused by closure (Figure 3B,D). Furthermore, the virus titer decreased significantly under the treatment of the TLR7 inhibitor and showed an increasing trend with the treatment of the TLR7 agonist (Figure 3E).

In addition, the activation of the Myd88 pathway regulated RABV-induced CCL2, and CXCL10 and IL-6 production was measured using the selective inhibitor of TLR7 and Myd88. BV-2 cells were pre-treated with the inhibitor of TLR7 (HY-124603) for 2 h and inhibitor of Myd88 (HY-50937) for 30 min. The results showed that pre-treatment with either HY-124603 or HY-50937 significantly inhibited CCL2 by approximately 71.21% (*p* < 0.01), CXCL10 by approximately 73.53% (*p* < 0.01) and IL-6 by approximately 73.63% (*p* < 0.01) (Figure 3F–H). Correspondingly, the dual inhibitors presented a stronger inhibitory effect on CCL2 (79.64%, *p* < 0.01), CXCL10 (80.03%, *p* < 0.01) and IL-6 (82.24%. *p* < 0.01) (Figure 3F–H). These data indicate that the Myd88 pathway is critically important in the RABV-induced production of CCL2, CXCL10 and IL-6 by the TLR7 signal.

### 2.4. Activation of NF-κB Pathway Regulates RABV-Induced Cytokine Expression via TLR7 Signal

Preliminary research indicates that TLR7 may serve as a crucial pattern recognition receptor for RABV, while the transcription factor NF-κB is pivotal in mediating the upregulation of chemokine gene expression with RABV infection [26]. To evaluate the dynamic relationship between TLR7 and NF-κB with RABV infection, BV-2 cells were pre-treated with the TLR7 inhibitor for 2 h. According to prior research, the translocation of NF-κB p65 to the nucleus exhibited a time-dependent increase following CVS-11 infection, peaking at 30 min post-infection [33]. Therefore, we measured the phosphorylation of NF-κB with CVS-11 infection at 30 min with or without the TLR7 inhibitor. It is considered that the degradation of IκBα leads to NF-κB dimer transport to the nucleus, so we also measured the level of IκBα in the cytoplasmic extract. As the results show, the phosphorylation of NF-κB p65 significantly decreased with the TLR7 inhibitor, and the TLR7 agonist enhanced the phosphorylation of NF-κB p65. Conversely, IκBα exhibited behavior that contrasted that of NF-κB p65, as the cytoplasmic degradation of IκBα was markedly accelerated in the presence of the TLR7 agonist (Figure 4A). Meanwhile, to represent the nuclear translocation of NF-κB p65 with CVS-11 infection via Image Stream analysis, BV-2 cells were pre-treated with or without the TLR7 inhibitor. Similar to the results above, we found that the nuclear translocation of NF-κB p65 showed a downward trend with the TLR7 inhibitor (Figure 4B). These results show that the phosphorylation and nuclear translocation of NF-κB p65 pre-treated with the TLR7 inhibitor coincided with IκBα degradation in the cytoplasm.

In addition, the activation of NF-κB pathway regulated RABV-induced CCL2, and CXCL10 and IL-6 production was measured using the selective inhibitor of TLR7 and NF-κB. BV-2 cells were pre-treated with the inhibitor of TLR7 (HY-124603) for 2 h and inhibitor of NF-κB (BAY11-7082) for 1 h. The results showed that pre-treatment with either HY-124603 or BAY11-7082 significantly inhibited the production of CCL2, CXCL10 and IL-6 (Figure 4C–E). Correspondingly, the dual inhibitors presented a stronger inhibitory effect on CCL2 (84.35%, *p* < 0.01), CXCL10 (86.55%, *p* < 0.01) and IL-6 (85.76%. *p* < 0.01) (Figure 4C–E). These data indicate that the NF-κB pathway is critically important in the RABV-induced production of CCL2, CXCL10 and IL-6 by the TLR7 signal.

## 3. Discussion

RABV is a highly virulent neurotropic pathogen that typically induces fatal CNS disorders. the infection process also produces an inflammatory response that causes damage to the target organ. Excessive inflammation often leads to nerve damage and CNS dysfunction [2]. In recent years, several studies have identified another role of inflammation in the development of rabies. The addition of inhibitors of related signaling pathways can reduce RABV load, decrease the expression of chemokines, and reduce the activation of microglial and intracranial inflammatory damage [36,37,38]. However, the mechanism of inflammation due to RABV infection, the role of inflammation in infection with RABV strains with different symptom types and pathogenicity, and the reasons for these differences require additional elucidation. In our study, we found that RABV infection leads to an inflammatory response, which is also accompanied by an increased expression of CCL2, CXCL10 and IL6. Inhibition of TLR7 markedly diminished the synthesis of these chemokines and cytokines, suggesting that TLR7 is essential for cytokine production induced by RABV. These findings were consistent with data reported in the primary investigation [34].

Viruses with different characteristics are recognized by different TLRs, which then trigger antiviral innate immunity. Since the discovery of TLRs, its role in the antiviral immune response has been gradually clarified. TLR2 and TLR4, expressed on cell membranes, participate in antiviral immune responses by recognizing products of infected cells, such as heat shock proteins [39]. TLR3, expressed on the endoplasmic reticulum and endosomal membrane, is able to recognize the double-stranded replicative intermediate of the RABV RNA. [40]. TLR7 expressed on the endosome membrane can bind single-stranded RNA [20]. TLR8, which is also distributed in endosomes, can recognize single-stranded RNA in human cells [41]. TLR9 can recognize the DNA of the virus [42]. TLRs are activated by binding to their corresponding ligand and can excite downstream signaling pathways. Different TLRs may activate the same signaling pathway via the same downstream splice molecule, such as splice protein Myd88. For rabies, Liu et al. found that antibody levels were elevated by using a TLR7 agonist as a vaccine adjuvant for RABV, suggesting that TLR7 played a critical role in RABV-induced neutralizing antibody production [26]. Li et al. reported that the pathogenicity of RABV was significantly increased in TLR7-deficient mice, but its specific effect and mechanism remained unclear [27]. Luo et al. demonstrated that TLR7 upregulated the expression of cytokines and chemokines related to humoral immunity, meanwhile contributing to RABV-induced antibody production by facilitating the formation of a germinal center and the gathering of germinal-center B cells [34]. In our study, we found that TLR7 was involved in the immunopathogenesis of infection with RABV, and the expression of CCL2, CXCL10 and IL-6 were significantly increased in the terminal stage of infection with RABV. These findings revealed a novel immunoregulatory mechanism of TLR7 in BV-2 cells infected by RABV.

TLR7 triggers the activation of a series of cytosolic adaptor molecules and specifically participates in the Myd88-independent signaling pathway [43]. Myd88-mediated signaling pathway plays a significant role in the TRL-driven immune response [44]. Myd88 can be considered the adaptor that recruits the serine-threonine kinase IRAK and TRAF6 to the TLR7 signaling pathway [45]. Furthermore, TRAF6 acts at the intersection of TLR7 signaling pathway, which leads to the activation of the NF-κB and Myd88 signaling pathway and induces the expression of downstream key nodes including IκBα and IRAK4. Our results showed the elevated of protein levels of the TLR7–Myd88 signaling pathway, including Myd88, IRAK4 and TRAF6 of BV-2 cells with RABV infection. In addition, blockade of TLR7 significantly reduced the expression of TRL7, Myd88, IRAK4 and TRAF6. Moreover, in concordance with the findings of previous studies demonstrating that TLR7 activation triggered the production of proinflammatory cytokines [46], a positive correlation between the expression of chemokines, cytokines and TRL7–Myd88-mediated signaling molecules was observed in our study.

The NF-κB transcription factor plays a pivotal role in the initiation and orchestration of both innate and adaptive immune responses, serving as the principal transcriptional regulator activated during TLR signaling pathways [47]. Our results demonstrated that the phosphorylation of NF-κB p65 significantly decreased with the TLR7 inhibitor and the TLR7 agonist enhanced the phosphorylation of NF-κB p65, indicating that the activation of the NF-κB signaling cascade via RABV stimulation is associated with TLR7. Conversely, IκBα exhibited results that were diametrically opposed to those of NF-κB p65, as the degradation of IκBα within the cytoplasmic compartment was markedly accelerated in response to the TLR7 agonist. Furthermore, our previous results showed that the nuclear translocation of NF-κB p65 in BV-2 cells was significantly increased after RABV stimulation [38]. However, the nuclear translocation of NF-κB p65 decreased after treatment with the NF-κB inhibitor. Inhibition of NF-κB with BAY-117082 and that of TLR7 with HY-124603 decreased the expression of CCL2, CXCL10 and IL-6 after RABV infection, suggesting that the influence of RABV on cytokine synthesis is modulated through the NF-κB signaling cascade via TLR7.

In conclusion, our research demonstrates that RABV stimulates the production of CCL2, CXCL10 and IL-6 in BV-2 microglial cells through the activation of TLR7, with the Myd88 and NF-κB signaling pathways playing a crucial role in this mechanism. Consequently, our findings may serve as a valuable reference and provide foundational experimental data for the clinical management of and in prevention strategies against rabies.

## 4. Materials and Methods

### 4.1. Mouse Studies

Six-week-old ICR mice were obtained from the Institute of Laboratory Animal Medicine at the Chinese Academy of Medical Sciences (CAMS & PUMC, Beijing, China). All procedures adhered to the Ministry of Health’s guidelines for Medical Laboratory Animals (1998) in China. To simulate RABV infection via a natural route, the mice (*n* ≥ 3 per group) were intramuscularly injected in the right thigh with RABV strains (5.6 × 10^3^ FFU; 25 μL in PBS), while mock controls received the same volume of sterile PBS. Disease progression was evaluated as described in our previous study [33]. Brains were then separately collected and stored according to testing requirements.

### 4.2. Cell Culture and Reagents

BV-2 cells were cultured in Dulbecco’s modified eagle medium (DMEM) with 10% fetal bovine serum (FBS) and maintained at 37 °C in a humidified incubator with 5% CO_2_. The TLR7 inhibitor (HY-124603, MCE, Princeton, NJ, USA), agonist (HY-117602, MCE, USA) and Myd88 inhibitor (HY-50937) were purchased from Med-Chem-Express. The NF-κB inhibitor (BAY11-7082, Sigma-Aldrich, St. Louis, MO, USA) was purchased from Sigma-Aldrich. Stocks were solubilized in DMSO and subsequently further diluted in a culture medium. The control group was administered DMSO at an equivalent concentration.

### 4.3. Virus Strains

This study utilized three strains of RABV, specifically SC16, HN10 and CVS-11. The RABV strains were preserved in our laboratory. SC16 is a field strain obtained from a rabid canine in Sichuan Province in 2006 [48]. HN10 is another field strain, isolated from a rabid human in Hunan Province in the same year [3]. The standard challenge virus, CVS-11, was supplied by the National Institutes for Food and Drug Control in China.

### 4.4. Viral Titration and Quantitative Real-Time PCR Assay (qRT-PCR)

BV-2 cells were seeded in 96-well plates and subsequently incubated with RABV at an MOI of 1, in a test medium for 1 h at 37 °C. Following incubation, the cells were washed and then cultured in complete medium. After a 72 h incubation period, the viral titer of BV-2 cells was assessed using a focus-forming assay as previously described [49]. Viral foci were detected using a FITC-conjugated anti-RABV N antibody (Fujirebio Diagnostics, Inc., Malvern, PA, USA). The foci were counted to determine the viral titer (FFU/mL). In addition, viral genome copies of RABV were quantified by Applied Biosystems™ QuantStudio™ 5 Real-Time PCR System (Applied Biosystems, Waltham, MA, USA).

### 4.5. Immunohistochemistry

Mouse brain tissue was fixed using formaldehyde, and IHC analysis was performed following the established protocol [3,50]. HE staining was employed to assess the histopathological alterations in the mouse brain [3,33,51]. Uninfected mice brain tissues served as the control group. The anti-TLR7 (no. 5632; Cell Signaling Technology; Beverly, MA, USA) antibody was used. DAB and hematoxylin were employed for the staining of tissue sections [3]. All immunohistochemical slides were evaluated and analyzed by the same pathologist.

### 4.6. ELISA

ELISA assays (R&D Systems; Wiesbaden-Nordenstadt, Germany) were employed to quantify the levels of CCL2, CXCL10 and IL-6 in BV-2 microglial cells infected with CVS-11, using non-infected cells as a control. To modulate the TLR7/Myd88/NF-κB signaling pathway, BV-2 cells were pre-treated for 2 h at 37 °C with the TLR7 inhibitor and agonist, for 30 min at 37 °C with the Myd88 inhibitor and for 1 h at 37 °C with the NF-κB inhibitor prior to exposure to CVS-11. The cell supernatant was subsequently harvested for ELISA analysis.

### 4.7. NF-κB Translocation

BV-2 cells were cultured onto six-well plates and infected with CVS-11 (MOI of 1) at 37 °C for 1 h. UV-inactivated (30 min) CVS-11 was used as a control. BV-2 cells were pretreated for 1 h at 37 °C with the NF-κB inhibitor before exposure to CVS-11. NF-κB translocation analysis was performed as previously described [33]. NF-Κb p65 (no. 8242; Cell Signaling Technology; Beverly, MA, USA) and Alexa Fluor 488-conjugated goat anti-rabbit IgG (no. 4412; Cell Signaling Technology; Beverly, MA, USA) were added for incubation after cell fixation. The Amnis Image Stream Mark II system was used to collect fluorescent images, and IDEAS 6.0 software (Amnis, Seattle, WA, USA) was used to examine the nuclear localization of transcription factors.

### 4.8. Western Blot

To investigate TLR7/Myd88/NF-κB signaling pathway via Western blotting, six-well plates were seeded with BV-2 cells. When the cell density reached 70% to 80%, different inhibitors and agonists were added to treat the cells. The inhibitor and agonist of TLR7 were pre-incubated for 2 h, the Myd88 inhibitor was pre-incubated for 30 min and the inhibitors of NF-κB were pre-incubated for 1 h. Following infection with CVS-11 for 1 h, the cells were cultured in medium for additional 30 min, 1 h, 2 h, 4 h, 6 h, 8 h, 12 h and 24 h. Cell sample collection and Western blot analysis were performed as previously described [33]. Antibodies against the following were used: TLR7 (no. 5632; Cell Signaling Technology; Beverly, MA, USA), Myd88 (no. 4283; Cell Signaling Technology; Beverly, MA, USA), IRAK4 (no. 4363; Cell Signaling Technology; Beverly, MA, USA), TRAF6 (no. 8028; Cell Signaling Technology; Beverly, MA, USA), NF-κB (no. 8242; Cell Signaling Technology; Beverly, MA, USA), IκBα (no. 4812; Cell Signaling Technology; Beverly, MA, USA), phospho-NF-κB (no. 3033; Cell Signaling Technology; Beverly, MA, USA) and β-actin (no. 4967; Cell Signaling Technology; Beverly, MA, USA). Chemiluminescent signals were detected using the Bio-Rad ChemiDoc™ XRS imaging system, and subsequent analysis of the blots was performed with Image Lab 3.0 software (Bio-Rad, Hercules, CA, USA).

### 4.9. Statistical Analysis

Data were processed using SPSS version 17.0 and GraphPad Prism software version 9.0, while the signal intensities of the Western blot bands were quantified utilizing ImageJ software 1.54j (National Institutes of Health, Bethesda, MD, USA). One-way analysis of variance (ANOVA) was employed to evaluate the data, with results deemed statistically significant at a threshold of *p* < 0.05 [52].

## 5. Conclusions

RABV strains of different virulence levels, both street and laboratory-selected strains, can significantly increase CCL2, CXCL10 and IL-6 productions via TLR7. The RABV stimulation of BV-2 cells activating the Myd88 pathway is TLR7-dependent. Inhibition of Myd88 or TLR7 activity decreases the expression of CCL2, CXCL10 and IL-6, which also reduces the activation and nuclear translocation of NF-κB p65 with RABV infection in BV-2 cells. Our study may provide a therapeutic strategy for RABV infection and protect neurons during viral infections.

## Figures and Tables

**Figure 1 ijms-25-09144-f001:**
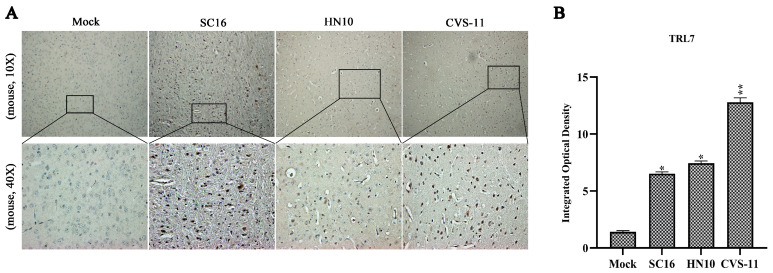
Infection with RABV strains exhibiting varying levels of virulence activating TLR7 in mouse brain tissue: (**A**) illustrative representations of IHC analysis of TLR7 in sections of mouse brains infected with the RABV strains SC16, HN10 and CVS-11; (**B**) integrated optical density of TLR7 in mouse brain tissue. Statistical evaluations were conducted utilizing ANOVA (* *p* < 0.05, ** *p* < 0.01).

**Figure 2 ijms-25-09144-f002:**
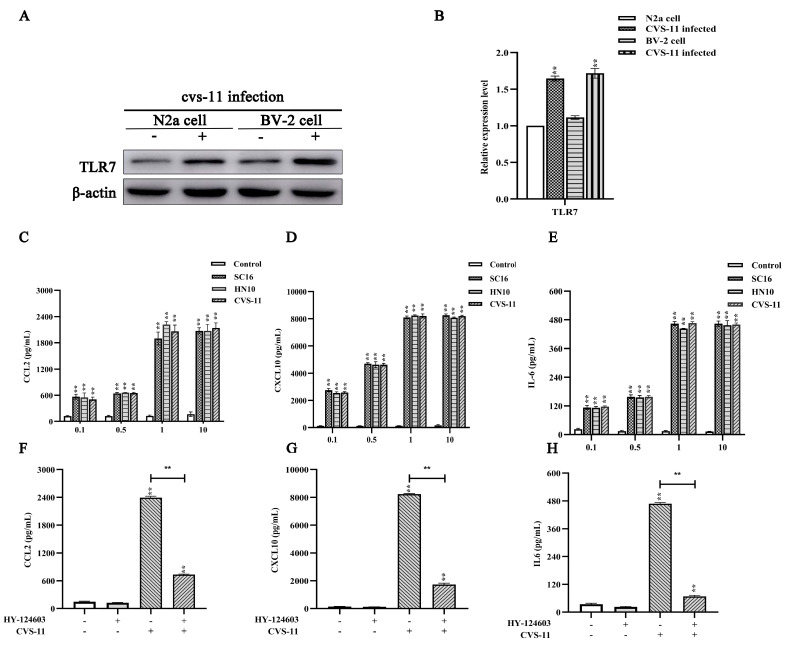
RABV inducing proinflammatory chemokines in vitro via TLR7. (**A**) N2a and BV-2 cells exposed to CVS-11, and subsequent Western blot analysis conducted to assess the expression of TLR7. (**B**) Quantitative assessment of the comparative signal intensities of TLR7. (**C**) Different RABV strains at an MOI of 0.1, 0.5, 1 and 10 of infected BV-2 cells. Cell culture supernatants were collected and the expression of CCL2 was measured. (**D**) The expression of CXCL10 of BV-2 cells infected with RABV at an MOI of 0.1, 0.5, 1 and 10. (**E**) The expression of IL-6 of BV-2 cells infected with RABV at an MOI of 0.1, 0.5, 1 and 10. (**F**) BV-2 cells incubated with or without the TLR7 inhibitor (HY-124603) before stimulation with CVS-11. The expression of CCL2 in the supernatant was measured via ELISA. (**G**) The expression of CXCL10 in the supernatant incubated with the TLR7 inhibitor (HY-124603) before stimulation with CVS-11. (**H**) The expression of IL-6 in the supernatant incubated with the TLR7 inhibitor (HY-124603) before stimulation with CVS-11. Statistical evaluations were conducted utilizing ANOVA (** *p* < 0.01).

**Figure 3 ijms-25-09144-f003:**
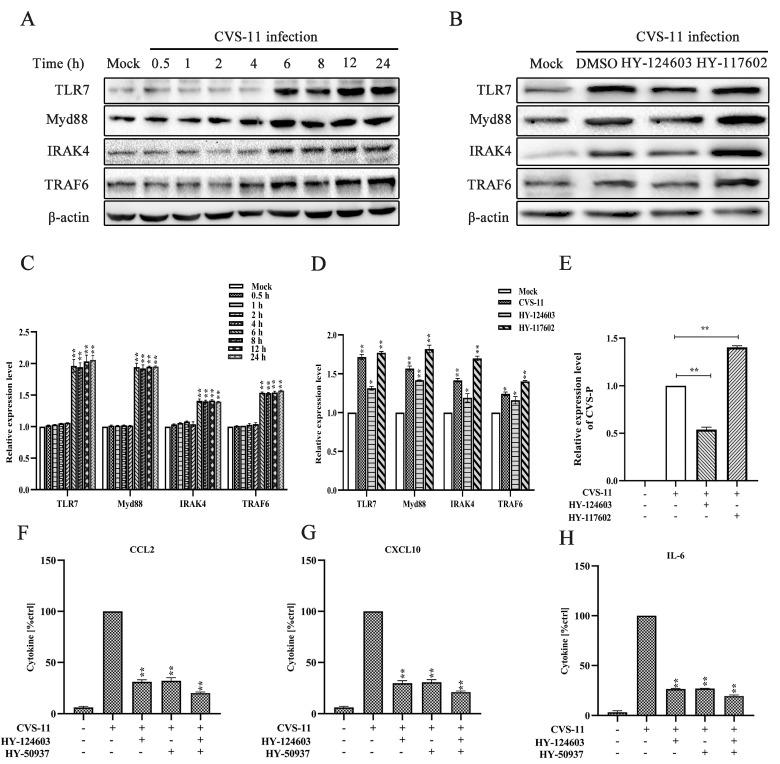
RABV inducing the activation of the Myd88 pathway through TLR7. (**A**) The expression levels of TLR7, MyD88, IRAK4 and TRAF6 assessed via Western blotting. (**B**) BV-2 cells pre-treated for 2 h with the inhibitor (HY-124603) and agonist (HY-117602) of TLR7 before stimulation with CVS-11 for 6 h. TRL7, Myd88, IRAK4 and TRAF6 were analyzed via Western blotting. (**C**) Quantitative assessment of the relative expression levels of TLR7, Myd88, IRAK4 and TRAF6, normalized against β-actin. (**D**) Quantitative assessment of the relative expression levels of TLR7, Myd88, IRAK4 and TRAF6 following normalization against β-actin, in response to treatment with TLR7 inhibitors (HY-124603) and agonists (HY-117602). (**E**) The expression levels of viral P gene mRNA, quantified using qRT-PCR following treatment with the TLR7 inhibitor (HY-124603) and agonist (HY-117602). (**F**) BV-2 cells pre-treated for 2 h with the TLR7 inhibitor (HY-124603) and for 30 min with the Myd88 inhibitor (HY-50937). The supernatants were collected, and the concentration of CCL2 was quantified at 24 h. (**G**) The expression of CXCL10 was measured at 24 h. (**H**) The expression of IL-6 was measured at 24 h. Statistical evaluations were conducted utilizing ANOVA (* *p* < 0.05, ** *p* < 0.01).

**Figure 4 ijms-25-09144-f004:**
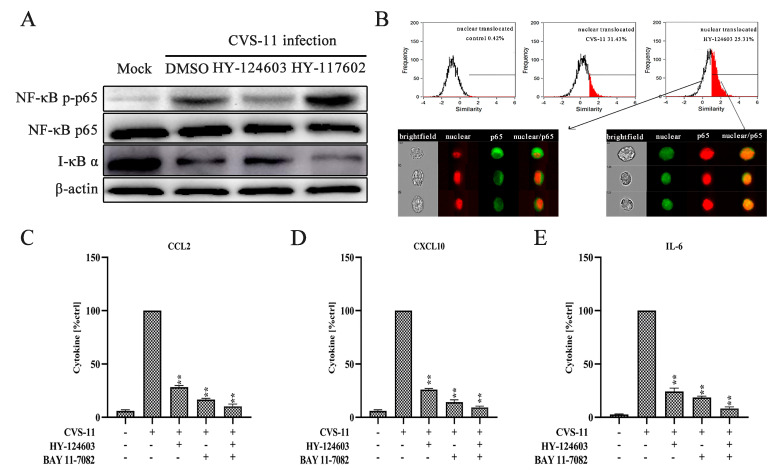
RABV inducing the phosphorylation and nuclear translocation of NF-κB. (**A**) BV-2 cells pre-treated for 2 h with the inhibitor (HY-124603) and agonist (HY-117602) of TLR7 before stimulation with CVS-11 for 30 min. NF-κB p-p65, NF-κB p65 and IκBα analyzed via Western blotting. (**B**) Analysis of the nuclear translocation of NF-κB in BV-2 cells incubated with or without the TLR7 inhibitor (HY-124603) with RABV infection for 30 min, carried out using Image Stream. (**C**) BV-2 cells under 2 h pre-treatment with the TLR7 inhibitor (HY-124603) and 1 h exposure to the NF-κB inhibitor (BAY11-7082). Afterward, the supernatants were collected, and CCL2 levels were quantified at 24 h. (**D**) The expression of CXCL10, measured at 24 h. (**E**) The expression of IL-6, measured at 24 h. Statistical evaluations were conducted utilizing ANOVA (** *p* < 0.01).

## Data Availability

The data and materials described in this manuscript are freely available.

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
