# Peer review of "Rabies Virus Regulates Inflammatory Response in BV-2 Cells through Activation of Myd88 and NF-κB Signaling Pathways via TLR7"

_ijms, 2024, doi:10.3390/ijms25179144_

Round 1

Reviewer 1 Report

Comments and Suggestions for Authors

General comments

In the study entitled “Rabies virus regulates the inflammatory response in BV-2 cells through activation of Myd88 and NF-κB signaling pathways via TLR7” the Authors’ purpose was to investigate the role of TLR7 in the immune response induced by rabies virus with different virulence. Authors showed that rabies virus with different virulence (SC16, HN10, CVS-11) significantly increased CCL2, CXCL10 and IL-6 production involving the Myd88 and the NF-κB pathways via TLR7 signal.

The information is of significant interest to the Journal's readers. The current study is well argued, and erudite, as well as being written to a good standard of English. It provides new insights on the topic. Some minor changes could improve the manuscript and some information should be added in the methods section.

Specific Comments

Please decrease the amount of wording duplication in the manuscript (Now it is of 49%, and appears to be too high)

The title as well as keywords accurately reflects the major findings of the work.

The abstract adequately summarize methodology, results, and significance of the study. Please indicate the statistical analysis applied as well as the P values when statistically significant differences were found

The introduction section is well written and it falls within the topic of the study. Punctuation should be improved. Material and methods are well written and meticulously describe methodology applied in the study.

Authors should indicate the specie-specificity of the kits or whether the kits have been previously validated. Moreover, Authors should indicate the intra- and inter-assay variability as well as the sensitivity of ELISA tests. Regarding statistical analysis, did Authors verify the normal distribution of data by a normality test before the application of One-way ANOVA (a parametric analysis)?

Please clarify this aspect.

Results are well written resulting clear for the reader. Authors well discuss the findings obtained in their study and appropriately justify them by referring to other studies available in literature.

The conclusion should be significantly improved as in the current form it is really short and not incisive. Authors should rewrite this section by summarizing the main findings gathered in the study, by emphasizing the significance of the study, and by clearly proposing new insights in the investigated field. 

Though Figures well represents the main findings of the study, they are very poor in quality. Please improve it for each figure.

Author Response

Comments 1: As part of our regular screening of all manuscripts, we checked and noticed that you mentioned human brain specimens in your manuscript. Manuscripts containing original descriptions of research conducted in humans must contain details of approval by a properly constituted research ethics committee. In order to continue processing your paper we must ask you to please provide the ethical approval information and blank informed consent form used in this study.

Response 1: Thank you for pointing this out. We agree with this comment. We have provided the necessary ethical documentation as attachments, which we have previously utilized in our earlier publications. This study focuses solely on mouse brain tissue and does not involve human brain tissue. Additionally, we have included the relevant animal ethics documentation as attachments. Thank you for your valuable feedback.

Comments 2: After iThenticate check for your manuscript, we found some sentences are similar with published papers (49%). In order to keep the novelty of your manuscript (less than 30% similarity is allowed), we kindly suggest you rephrase the highlighted sentences in your manuscript (find in the attachment - plagiarism report and the original paper in which you need to make changes). Please revise it during revision.

Response 2: Thank you for pointing this out. We agree with this comment. We have revised the full text and reduced the repetition rate based on the attachment - plagiarism report and the original paper. Thank you for your valuable feedback.

Comments 3: This is very interesting research manuscript; the authors investigated the role of TLR7-mediated inflammation via activation of Myd88 and NF-kB upon rabies virus infection. The authors found that rabies virus induces inflammation via TLR7 pathway dependent manner. The manuscript shows well-designed experiments including appropriate controls and scientific rigor. this reviewer did not find any weakness in the study.

The objective of this manuscript is to evaluate the role of toll-like receptor 7 in the rabies virus-mediated inflammation. The manuscript provides a new mechanistic insight of the importance of TLR7 in the inflammation trigger by rabies virus infection.

Whereas other reports show the critical role of MAPK and NFkB pathway in the inflammatory response to rabies virus, this study evaluates the inflammatory response from TLR perspective which provide new knowledge to the filed.

The authors may consider evaluating the inflammatory response in the brain and evaluate the brain damage caused by the inflammation.

The conclusions are appropriate and consistent with the main question.

The references are appropriate.

Figures and Tables: The manuscript is not well presented, the figures should be presented after each result, it was difficult to find the figure after reading the each result section.

Response 3: Thank you for pointing this out. We agree with this comment. We have adjusted the resolution of all images to maintain their quality and re-uploaded them individually. Thank you for your valuable feedback.

Comments 4: In the study entitled “Rabies virus regulates the inflammatory response in BV-2 cells through activation of Myd88 and NF-κB signaling pathways via TLR7” the Authors’ purpose was to investigate the role of TLR7 in the immune response induced by rabies virus with different virulence. Authors showed that rabies virus with different virulence (SC16, HN10, CVS-11) significantly increased CCL2, CXCL10 and IL-6 production involving the Myd88 and the NF-κB pathways via TLR7 signal.

The information is of significant interest to the Journal's readers. The current study is well argued, and erudite, as well as being written to a good standard of English. It provides new insights on the topic. Some minor changes could improve the manuscript and some information should be added in the methods section.

Response 4: Thank you for pointing this out. We agree with this comment. We have revised all parts of the article based on the attachment - plagiarism report and the original paper. Thank you for your valuable feedback.

Comments 5: Please decrease the amount of wording duplication in the manuscript (Now it is of 49%, and appears to be too high)

Response 5: Thank you for pointing this out. We agree with this comment. We have reduced the repetition rate based on the attachment - plagiarism report and the original paper. Thank you for your valuable feedback.

Comments 6: The title as well as keywords accurately reflects the major findings of the work.

The abstract adequately summarize methodology, results, and significance of the study. Please indicate the statistical analysis applied as well as the P values when statistically significant differences were found.

Response 6: Thank you for pointing this out. We agree with this comment. We have described the analytical methods in the abstract and indicated the P-values of the significant differences. Specific modifications are as follows: “Rabies is a fatal neurological infectious disease caused by rabies virus (RABV) which invades the central nervous system (CNS). RABV with varyingdifferent virulence regulates chemokine ex-pression and signaling pathways activation for immune responses remains to be elucidated. The relationship between toll-like receptors (TLRs) and immune response induced by RABV has not been fully clarified. In this studyHere, we investigated the role of TLR7 in the immune response induced by RABV and one-way analysis of variance (ANOVA) was employed to evaluate the data . We found that RABV with different virulence (SC16, HN10, CVS-11) significantly increased CCL2, CXCL10 and IL-6 production. Blocking assays indicated that the TLR7 inhibitor reduced the ex-pression of CCL2, CXCL10 and IL-6 (P<0.01). RABV stimulation of BV-2 cells activated the Myd88 pathway was TLR7-dependent, whereas inhibition of Myd88 activity reduced the expression of CCL2, CXCL10 and IL-6 (P<0.01). Meanwhile, RABV stimulation of BV-2 cells resulted in TRL7-mediated activation of NF-κB and induced nuclear translocation of NF-κB p65. CCL2, CXCL10 and IL-6 release was attenuated by the specific NF-κB inhibitor (P<0.01). The findings above demonstrate that RABV induced expression of CCL2, CXCL10 and IL-6 involves Myd88 and NF-κB pathways via TLR7 signal.”

Comments 7: The introduction section is well written and it falls within the topic of the study. Punctuation should be improved. Material and methods are well written and meticulously describe methodology applied in the study.

Response 7: Thank you for pointing this out. We agree with this comment. We have improved the punctuation of the introduction. Thank you for your valuable feedback.

Comments 8: Authors should indicate the specie-specificity of the kits or whether the kits have been previously validated. Moreover, Authors should indicate the intra- and inter-assay variability as well as the sensitivity of ELISA tests. Regarding statistical analysis, did Authors verify the normal distribution of data by a normality test before the application of One-way ANOVA (a parametric analysis)?

Response 8: Thank you for pointing this out. We agree with this comment. The specie-specificity, the intra- and inter-assay variability and the sensitivity of ELISA kit has been verified in our previous experiments. In addition, a normal distribution test was performed on the data before one-way ANOVA, and our data fit the normal distribution. Thank you for your valuable feedback.

Comments 9: Results are well written resulting clear for the reader. Authors well discuss the findings obtained in their study and appropriately justify them by referring to other studies available in literature.

The conclusion should be significantly improved as in the current form it is really short and not incisive. Authors should rewrite this section by summarizing the main findings gathered in the study, by emphasizing the significance of the study, and by clearly proposing new insights in the investigated field.

Though Figures well represents the main findings of the study, they are very poor in quality. Please improve it for each figure.

Response 9: Thank you for pointing this out. We agree with this comment. We have revised our conclusion as “RABV strains of different virulence, both street and laboratory-fixed strains, can significantly increase CCL2, CXCL10 and IL-6 productions via TLR7. RABV stimulation of BV-2 cells activated the Myd88 pathway is TLR7-dependent. Inhibition of Myd88 or TLR7 activity decrease the expression of CCL2, CXCL10 and IL-6, which also reduce the activation and nuclear translocation of NF-κB p65 upon RABV infection in BV-2 cells. Our study may provide a therapeutic strategy for RABV infection and protect neurons during viral infections”. In addition, we have adjusted the resolution of figures and uploaded them separately. Thank you for your valuable feedback.

Reviewer 2 Report

Comments and Suggestions for Authors

This is very interesting research manuscript; the authors investigated the role of TLR7-mediated inflammation via activation of Myd88 and NF-kB upon rabies virus infection. The authors found that rabies virus induces inflammation via TLR7 pathway dependent manner. The manuscript shows well-designed experiments including appropriate controls and scientific rigor. this reviewer did not find any weakness in the study.

The objective of this manuscript is to evaluate the role of toll-like receptor 7 in the rabies virus-mediated inflammation. The manuscript provides a new mechanistic insight of the importance of TLR7 in the inflammation trigger by rabies virus infection.

Whereas other reports show the critical role of MAPK and NFkB pathway in the inflammatory response to rabies virus, this study evaluates the inflammatory response from TLR perspective which provide new knowledge to the filed.

The authors may consider evaluating the inflammatory response in the brain and evaluate the brain damage caused by the inflammation.

The conclusions are appropriate and consistent with the main question.

The references are appropriate.

Figures and Tables: The manuscript is not well presented, the figures should be presented after each result, it was difficult to find the figure after reading the each result section.

Author Response

(The authors gave the same response as above.)
